# DEEP IMITATIVE MODELS FOR FLEXIBLE INFERENCE, PLANNING, AND CONTROL

## ABSTRACT

Imitation learning provides an appealing framework for autonomous control: in many tasks, demonstrations of preferred behavior can be readily obtained from human experts, removing the need for costly and potentially dangerous online data collection in the real world. However, policies learned with imitation learning have limited flexibility to accommodate varied goals at test time. Model-based reinforcement learning (MBRL) offers considerably more flexibility, since a predictive model learned from data can be used to achieve various goals at test time. However, MBRL suffers from two shortcomings. First, the model does not help to choose desired or safe outcomes – its dynamics estimate only what is possible, not what is preferred. Second, MBRL typically requires additional online data collection to ensure that the model is accurate in those situations that are actually encountered when attempting to achieve test time goals. Collecting this data with a partially trained model can be dangerous and time-consuming. In this paper, we aim to combine the benefits of imitation learning and MBRL, and propose imitative models: probabilistic predictive models able to plan expert-like trajectories to achieve arbitrary goals. We find this method substantially outperforms both direct imitation and MBRL in a simulated autonomous driving task, and can be learned efficiently from a fixed set of expert demonstrations without additional online data collection. We also show our model can flexibly incorporate user-supplied costs at test-time, can plan to sequences of goals, and can even perform well with imprecise goals, including goals on the wrong side of the road.

## 1 Introduction

Reinforcement learning (RL) algorithms offer the promise of automatically learning behaviors from raw sensory inputs with minimal engineering. However, RL generally requires *online* learning: the agent must collect more data with its latest strategy, use this data to update a model, and repeat. While this is natural in some settings, deploying a partially-trained policy on a real-world autonomous system, such as a car or robot, can be dangerous. In these settings the behavior must be learned *offline*, usually with expert demonstrations. How can we incorporate such demonstrations into a flexible robotic system, like an autonomous car? One option is imitation learning (IL), which can learn policies that stay near the expert's distribution. Another option is model-based RL (MBRL) (Kuvayev & Sutton, 1996; Deisenroth & Rasmussen, 2011), which can use the data to fit a dynamics model, and can in principle be used with planning algorithms to achieve any user-specified goal at test time. However, in practice, model-based and model-free RL algorithms are vulnerable to distributional drift (Thrun, 1995; Ross & Bagnell, 2010): when acting according to the learned model or policy, the agent visits states different from those seen during training, and in those it is unlikely to determine an effective course of action. This is especially problematic when the data intentionally excludes adverse events, such as crashes. A model ignorant to the possibility of a crash cannot know how to prevent it. Therefore, MBRL algorithms usually require online collection and training (Englert et al., 2013; Liang et al., 2018). Imitation learning algorithms use expert demonstration data and, despite similar drift shortcomings (Ross et al., 2011), can sometimes learn effective policies without additional online data collection (Zhang et al., 2018). However, standard IL offers little task flexibility since it only predicts low-level behavior. While several works augmented IL with goal conditioning (Dosovitskiy & Koltun, 2016; Codevilla et al., 2018), these goals must be specified in advance during training, and are typically simple (*e.g.*, turning left or right).

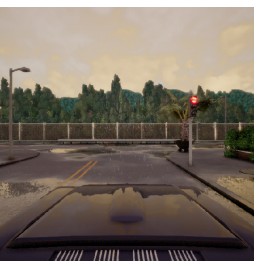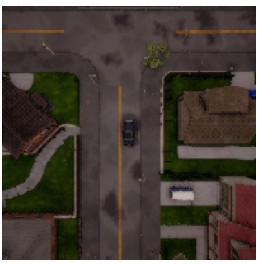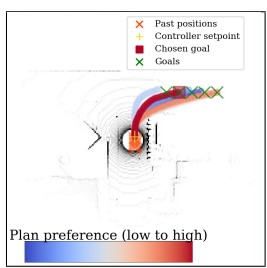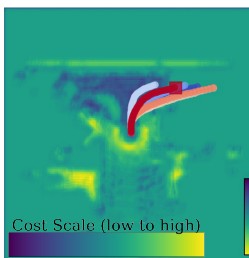

Figure 1: We apply our approach to navigation in CARLA (Dosovitskiy et al., 2017). *Columns 1,2:* Images depicting the current scene. The overhead image depicts a $50\,\mathrm{m}^2$ area. *Column 3:* LIDAR input and goals are provided to our deep imitative trajectory model, and plans to the goals are computed under the model's likelihood objective, and colored according to their ranking under the objective, with red indicating the best plan. The red square indicates the chosen high-level goal, and the yellow cross indicates a point along our plan used as a setpoint for a PID controller. The LIDAR map is $100\,\mathrm{m}^2$, and each goal is $\geq 20\,\mathrm{m}$ away from the vehicle. *Column 4:* Our model can incorporate arbitrary test-time costs, and use them to adjust its planning objective and plan ranking.

The goal in our work is to devise a new algorithm that combines the advantages of IL and MBRL, affording both the flexibility to achieve new user-specified goals at test time and the ability to learn entirely from offline data. By learning a deep probabilistic predictive model from expert-provided data, we capture the distribution of expert behaviors without using manually designed reward functions. To plan to a goal, our method infers the most probable expert state trajectory, conditioned on the current position and reaching the goal. By incorporating a model-based representation, our method can easily plan to previously unseen user-specified goals while respecting rules of the road, and can be flexibly repurposed to perform a wide range of test-time tasks without any additional training. Inference with this model resembles trajectory optimization in model-based reinforcement learning, and learning this model resembles imitation learning.

Our method's relationship to other work is illustrated in Fig. 2. We demonstrate our method on a simulated autonomous driving task (see

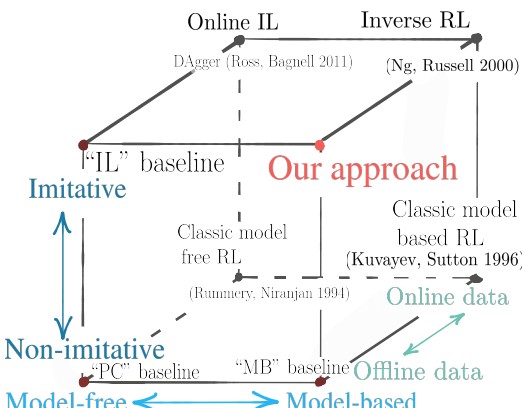

Figure 2: A brief taxonomy of learning-based control methods. In our scenario, we avoid online data collection, specifically from the policy we seek to imitate. We structure our imitation learner with a model to make it flexible to new tasks at test time. We compare against other offline approaches (front face).

Fig. 1). A high-level route planner provides navigational goals, which our model uses to automatically generate plans that obey the rules of the road, inferred entirely from data. In contrast to IL, our method produces an interpretable distribution over trajectories and can follow a variety of goals without additional training. In contrast to MBRL, our method generates human-like behaviors without additional data collection or learning. In our experiments, our approach substantially outperforms both MBRL and IL: it can efficiently learn near-perfect driving through the static-world CARLA simulator from just 7,000 trajectories obtained from 19 hours of driving. We also show that our model can flexibly incorporate and achieve goals not seen during training, and is robust to errors in the high-level navigation system, even when the high-level goals are on the wrong side of the road. Videos of our results are available.[1]

---

[1] https://sites.google.com/view/imitativeforecastingcontrol

## 2 Deep Imitative Models

To learn robot dynamics that are not only possible, but preferred, we construct a model of expert behavior. We fit a probabilistic model of trajectories, $q$, to samples of expert trajectories drawn from an unknown distribution $p$. A probabilistic model is necessary because expert behavior is often stochastic and multimodal: *e.g.*, choosing to turn either left or right at an intersection are both common decisions. Because an expert's behavior depends on their perception, we condition our model, $q$, on observations $\phi$. In our application, $\phi$ includes LIDAR features $\chi \in \mathbb{R}^{H \times W \times C}$ and a small window of previous positions $s_{-\tau:0} = \{s_{-\tau}, \ldots, s_0\}$, such that $\phi = \{\chi, s_{-\tau:0}\}$.

By training $q(s_{1:T}|\phi)$ to forecast expert trajectories with high likelihood, we model the scene-conditioned expert dynamics, which can score trajectories by how likely they are to come from the expert. At test time, $q(s_{1:T}|\phi)$ serves as a learned prior over the set of *undirected* expert trajectories. To execute samples from this distribution is to imitate an expert driver in an undirected fashion. We first describe how we use the generic form of this model to plan, and then discuss our particular implementation in Section 2.2.

### 2.1 Imitative Planning to Goals

Besides simply imitating the expert demonstrations, we wish to *direct* our agent to desired goals at test time, and have the agent reason automatically about the mid-level details necessary to achieve these goals. In general, we can define a driving task by a set of goal variables $\mathcal{G}$. We will instantiate examples of $\mathcal{G}$ concretely after the generic goal planning derivation. The probability of a plan conditioned on the goal $\mathcal{G}$ is given as posterior distribution $p(s_{1:T}|\mathcal{G}, \phi)$. Planning a trajectory under this posterior corresponds to MAP inference with prior $q(s_{1:T}|\phi)$ and likelihood $p(\mathcal{G}|s_{1:T}, \phi)$. We briefly derive the MAP inference result starting from the posterior maximization objective, which uses the learned Imitative Model to generate plans that achieve abstract goals:

$$
\begin{aligned}
\max_{s_{1:T}} \mathcal{L}(s_{1:T}, \mathcal{G}, \phi) &= \max_{s_{1:T}} \log p(s_{1:T}|\mathcal{G}, \phi) \\
&= \max_{s_{1:T}} \log q(s_{1:T}|\phi) + \log p(\mathcal{G}|s_{1:T}, \phi) - \log p(\mathcal{G}|\phi) \\
&= \max_{s_{1:T}} \log \underbrace{q(s_{1:T}|\phi)}_{\text{imitation prior}} + \log \underbrace{p(\mathcal{G}|s_{1:T}, \phi)}_{\text{goal likelihood}}.
\end{aligned}
\tag{1}
$$

**Waypoint planning**: One example of a concrete inference task is to plan towards a specific goal location, or waypoint. We can achieve this task by using a tightly-distributed goal likelihood function centered at the user's desired final state. This effectively treats a desired goal location, $g_T$, as if it were a noisy observation of a future state, with likelihood $p(\mathcal{G}|s_{1:T}, \phi) = \mathcal{N}(g_T|s_T, \epsilon I)$. The resulting inference corresponds to planning the trajectory $s_{1:T}$ to a likely point under the distribution $\mathcal{N}(g_T|s_T, \epsilon I)$. We can also plan to successive states with $\mathcal{G} = (g_{T-K}, \ldots, g_T)$ with goal likelihood $p(\mathcal{G}|s_{1:T}, \phi) = \prod_{k=T-K}^{T} \mathcal{N}(g_k|s_k, \epsilon I)$ if the user (or program) wishes to specify the desired end velocity or acceleration when reached the final goal $g_T$ location (Fig. 3). Alternatively, a route planner may propose a set of waypoints with the intention that the robot should reach any one of them. This is possible using a Gaussian mixture likelihood and can be useful if some of those waypoints along a route are inadvertently located at obstacles or potholes (Fig. 4).

Waypoint planning leverages the advantage of conditional imitation learning: a user or program can communicate *where* they desire the agent to go without knowing the best and safest actions. The planning-as-inference procedure produces paths similar to how an expert would acted to reach the given goal. In contrast to black-box, model-free conditional imitation learning that regresses controls, our method produces an explicit plan, accompanied by an explicit score of the plan's quality. This provides both interpretability and an estimate of the feasibility of the plan.

**Costed planning**: If the user desires more control over the plan, our model has the additional flexibility to accept arbitrary user-specified costs $c$ at test time. For example, we may have updated knowledge of new hazards at test time, such as a given map of potholes (Fig. 4) or a predicted cost map. Given costs $c(s_i|\phi)$, this can be treated by including an optimality variable $\mathcal{C}$ in $\mathcal{G}$, where $p(\mathcal{C} = 1|s_{1:T}, \phi) \propto \prod_{t=1}^{T} \exp -c(s_t|\phi)$ (Todorov, 2007; Levine, 2018). The goal log-likelihood is $\log p(\{g_T, \mathcal{C} = 1\}|s_{1:T}, \phi) = \log \mathcal{N}(g_T|s_T, \epsilon I) + \sum_{t=1}^{N} -c(s_t|\phi)$.

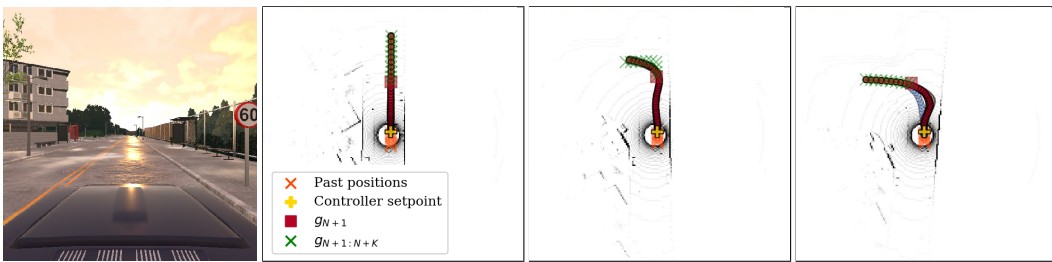

Figure 3: Planning to a sequence of goals (here, 10) allows for more control over the inferred paths.

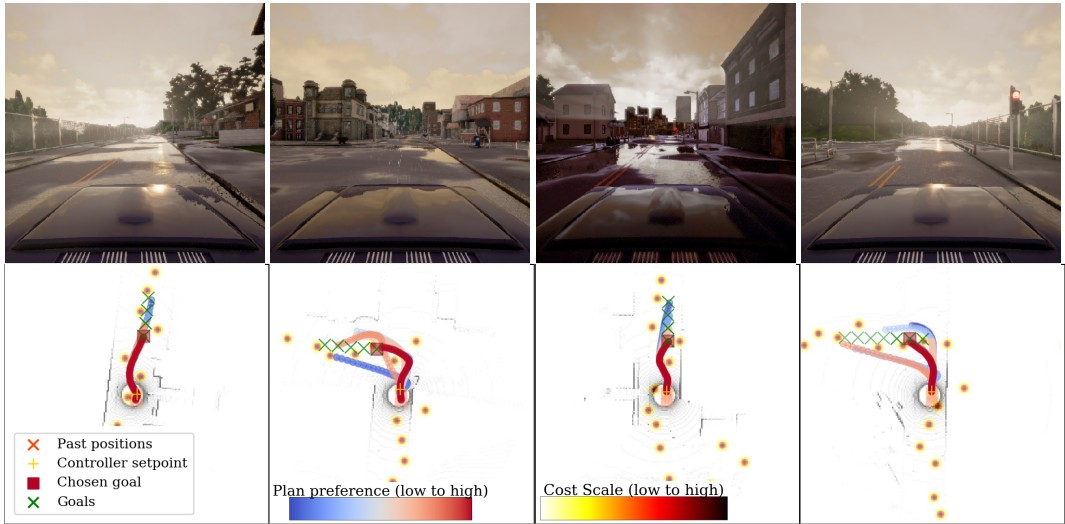

Figure 4: Imitative planning to goals subject to a cost at test time. The cost bumps corresponds to simulated "potholes," which the imitative planner is tasked with avoiding. The imitative planner generates and prefers routes that curve around the potholes, stay on the road, and respect intersections. Demonstrations of this behavior were never observed by our model.

## 2.2 Model Implementation

The primary structural requirement of an Imitative Model is the ability to compute $q(s_{1:T}|\phi)$. The ability to also compute gradients $\nabla_{s_{1:T}} q(s_{1:T}|\phi)$ enables gradient-based optimization for planning. Finally, the quality and efficiency of learning are important. One deep generative model for Imitation Learning is the Reparameterized Pushforward Policy (R2P2) (Rhinehart et al., 2018). R2P2's use of *pushforward distributions* (McCann et al., 1995), employed in other invertible generative models (Rezende & Mohamed, 2015; Dinh et al., 2016) allows it to efficiently minimize both false positives and false negatives (type I and type II errors) (Neyman & Pearson, 1933). Optimization of $KL(p, q)$, which penalizes mode loss (false negatives), is straightforward with R2P2, as it can evaluate $q(s_{1:T}|\phi)$. Here, $p$ is the sampleable, but unknown, distribution of expert behavior. Reducing false positives corresponds to minimizing $KL(q, p)$, which penalizes $q$ heavily for generating bad

---

**Algorithm 1** IMITATIVEPLAN($q_\theta, \mathcal{G}, \phi$)

1: Define MAP objective $\mathcal{L}$ with $q_\theta$ according to Eq. 1          ▷ Incorporate the Imitative Model
2: Initialize $s_{1:T}$
3: **while** not converged **do**          ▷ Approx. MAP inference
4:      $s_{1:T} \leftarrow s_{1:T} + \nabla_{s_{1:T}} \mathcal{L}(s_{1:T}, \mathcal{G}, \phi)$
5: **end while**
6: **return** $s_{1:T}$

---

samples under $p$. As $p$ is unknown, R2P2 first uses a spatial cost model $\tilde{p}$ to approximate $p$, which we can also use as $c$ in our planner. The learning objective is $KL(p, q) + \beta KL(q, \tilde{p})$.

In R2P2, $q(s_{1:T}|\phi)$ is induced by an invertible, differentiable function: $f(z; \phi) : \mathbb{R}^{2T} \mapsto \mathbb{R}^{2T}$, which warps latent samples from a base distribution $z \sim q_0 = \mathcal{N}(0, I_{2T \times 2T})$ to the output space over $s_{1:T}$. $f$ embeds the evolution of learned discrete-time stochastic dynamics; each state is given by:

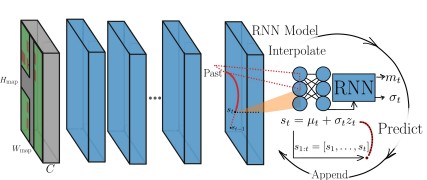

$$s_t = \underbrace{s_{t-1} + (s_{t-1} - s_{t-2}) + m_t(s_{1:t-1}, \phi)}_{\mu_t(s_{1:t-1}, \phi)} + \sigma_t(s_{1:t-1}, \phi) z_t.$$

Figure 5: Architecture of $m_t$ and $\sigma_t$, modified from (Rhinehart et al., 2018) with permission.

The $m_t \in \mathbb{R}^2$ and $\sigma_t \in \mathbb{R}^{2 \times 2}$ are computed by expressive, nonlinear neural networks that observe previous states and LIDAR input. The resulting trajectory distribution is complex and multimodal. We modified the RNN method described by Rhinehart et al. (2018) and used LIDAR features $\chi = \mathbb{R}^{200 \times 200 \times 2}$, with $\chi_{ij}$ representing a 2-bin histogram of points below and above the ground in $0.5\,\mathrm{m}^2$ cells (Fig 5). We used $T = 40$ trajectories at 5Hz (8 seconds of prediction or planning), $\tau = 19$.

## 2.3 Imitative Driving

At test time, we use three layers of spatial abstractions to plan to a faraway destination, common to model-based (not end-to-end) autonomous vehicle setups: coarse route planning over a road map, path planning within the observable space, and feedback control to follow the planned path (Paden et al., 2016; Schwarting et al., 2018). For instance, a route planner based on a conventional GPS-based navigation system might output waypoints at a resolution of 20 meters – roughly indicating the direction of travel, but not accounting for the rules of the road or obstacles. The waypoints are treated as goals and passed to the Imitative Planner (Algorithm 1), which then generates a path chosen according to the optimization in Eq. 1. These plans are fed to a low-level controller (we use a PID-controller) that follows the plan. In Fig. 6 we illustrate how we use our model in our application.

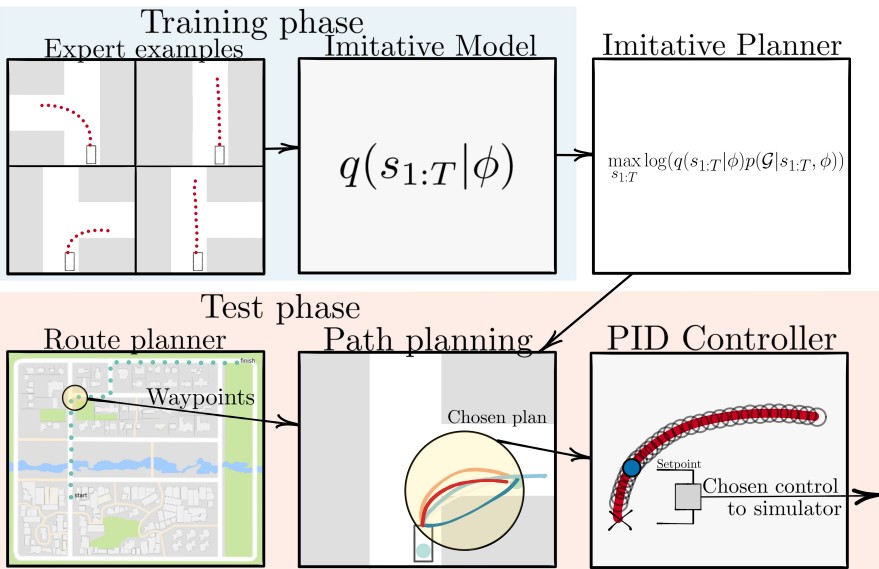

Figure 6: Illustration of our method applied to autonomous driving. Our method trains an Imitative Model from a dataset of expert examples. After training, the model is repurposed as an Imitative Planner. At test time, a route planner provides waypoints to the Imitative Planner, which computes expert-like paths to each goal. The best plan chosen according to the planning objective, and provided to a low-level PID-controller in order to produce steering and throttle actions.

## 3    Related Work

Previous work has explored conditional IL for autonomous driving. Two model-free approaches were proposed by Codevilla et al. (2018), to map images to actions. The first uses three network "heads", each head only trained on an expert's left/straight/right turn maneuvers. The robot is directed by a route planner that chooses the desired head. Their second method input the goal location into the network, however, this did not perform as well. While model-free conditional IL can be effective given a discrete set of user directives, our model-based conditional IL has several advantages. Our model has flexibility to handle more complex directives post training, *e.g.* avoiding hazardous potholes (Fig. 4) or other costs, the ability to rank plans and goals by its objective, and interpretability: it can generate entire planned and unplanned (undirected) trajectories.

Work by Liang et al. (2018) also uses multi-headed model-free conditional imitation learning to "warm start" a DDPG driving algorithm (Lillicrap et al., 2015). While warm starting hastens DDPG training, any subsequent DDPG post fine-tuning is inherently trial-and-error based, without guarantees of safety, and may crash during this learning phase. By contrast, our method never executes unlikely transitions w.r.t. expert behavior at training time nor at test time. Our method can also stop the car if no plan reaches a minimum threshold, indicating none are likely safe to execute.

While our target setting is offline data collection, online imitation learning is an active area of research in the case of hybrid IL-RL (Ross & Bagnell, 2014; Sun et al., 2018) and "safe" IL (Sun et al., 2017; Menda et al., 2017; Zhang & Cho, 2017). Although our work does not consider multiagent environments, several methods predict the behavior of other vehicles or pedestrians. Typically this involves recurrent neural networks combined with Gaussian density layers or generative models based on some context inputs such as LIDAR, images, or known positions of external agents (Lee et al., 2017; Schmerling et al., 2018; Zyner et al., 2018; Gupta et al., 2018; Ma et al., 2017). However, none of these methods can evaluate the likelihood of trajectories or repurpose their model to perform other inference tasks. Other methods include inverse reinforcement learning to fit a probabilistic reward model to human demonstrations using the principle of maximum entropy (Ziebart et al., 2008; Sadigh et al., 2016; Rhinehart & Kitani, 2017).

## 4    Experiments

We evaluate our method using the CARLA urban driving simulator (Dosovitskiy et al., 2017). Each test episode begins with the vehicle randomly positioned on a road in the `Town01` or `Town02` maps. The task is to drive to a goal location, chosen to be the furthest road location from the vehicle's initial position. As shown in Fig. 6, we use three layers of spatial abstractions to plan to the goal location, common to model-based (not end-to-end) autonomous vehicle setups: coarse route planning over a road map, path planning within the observable space, and feedback control to follow the planned path (Paden et al., 2016; Schwarting et al., 2018). First, we compute a route to the goal location using A* given knowledge of the road graph. Second, we set waypoints along the route no closer than $20\,\mathrm{m}$ of the vehicle at any time to direct the vehicle. Finally, we use a PID-controller to compute the vehicle steering value. The PID-controller was tuned to steer the vehicle towards a setpoint (target) 5 meters away along the planned path.

We consider four metrics for this task: 1) Success rate in driving to the goal location without any collisions. 2) Proportion of time spent driving in the correct lane. 3) Frequency of crashes into obstacles. 4) Passenger comfort, by comparing the distribution of accelerations (and higher-order terms) between each method. To contrast the benefits of our method against existing approaches, we compare against several baselines that all receive the same inputs and training data as our method. Since our approach bridges model-free IL and MBRL, we include an IL baseline algorithm, and a MBRL baseline algorithm.

**PID control:** The PID baseline uses the PID-controller to follow the high-level waypoints along the route. This corresponds to removing the middle layer of autonomous vehicle decision abstraction, which serves as a baseline for the other methods. The PID controller is effective when the setpoint is several meters away, but fails when the setpoint is further away (*i.e.* at $20\,\mathrm{m}$), causing the vehicle to cut corners at intersections.

**Conditional Imitation Learning:** We designed an IL baseline to control the vehicle. A common straightforward approach to IL is behavior-cloning: learning to predict the actions taken by a demon-

strator (Pomerleau, 1989; Bojarski et al., 2016; Mahler & Goldberg, 2017; Codevilla et al., 2018). Our setting is that of goal-conditioned IL: in order to achieve different behaviors, the imitator is tasked with generating controls after observing a target high-level waypoint and $\phi$. We designed two baselines: one with the `branched` architecture of Codevilla et al. (2018), where actions are predicted based on left/straight/right "commands" derived from the waypoints, and other that predicts the setpoint for the PID-controller. Each receives the same $\phi$ and is trained with the same set of trajectories as our main method. We found the latter method very effective for stable control on straightaways. When the model encounters corners, however, prediction is more difficult, as in order to successfully avoid the curbs, the model must implicitly plan a safe path. In the latter method, we used a network architecture nearly identical to our approach's..

**Model-based RL:** To compare against a purely model-based reinforcement learning algorithm, we propose a model-predictive control baseline. This baseline first learns a forwards dynamics model $f : (s_{t-3}, s_{t-2}, s_{t-1}, s_t, a_t) \rightarrow s_{t+1}$ given observed expert data ($a_t$ are recorded vehicle actions). We use an MLP with two hidden layers, each 100 units. Note that our forwards dynamics model does not imitate the expert preferred actions, but only models what is physically possible. Together with the same LIDAR map $\chi$ our method uses to locate obstacles, this baseline uses its dynamics model to plan a reachability tree (LaValle, 2006) through the free-space to the waypoint while avoiding obstacles. We plan forwards over 20 time steps using a breadth-first search search over CARLA steering angle $\{-0.3, -0.1, 0., 0.1, 0.3\}$, noting valid steering angles are normalized to $[-1, 1]$, with constant throttle at 0.5, noting the valid throttle range is $[0, 1]$. Our search expands each state node by the available actions and retains the 50 closest nodes to the waypoint. The planned trajectory efficiently reaches the waypoint, and can successfully plan around perceived obstacles to avoid getting stuck. To convert the LIDAR images into obstacle maps, we expanded all obstacles by the approximate radius of the car, 1.5 meters.

Performance results that compare our methods against baselines according to multiple metrics are includes in Table 1. With the exception of the success rate metric, lower numbers are better. We define success rate as the proportion of episodes where the vehicles navigated across the road map to a goal location on the other side without any collisions. In our experiments we do not include any other drivers or pedestrians, so a collision is w.r.t. a stationary obstacle. Collision impulse (in $N \cdot s$) is the average cumulative collision intensities over episodes. "Wrong lane" and "Off road" percentage of the vehicle invading other lanes or offroad (averaged over time and episodes). While safety metrics are arguably the most important metric, passenger comfort is also relevant. Passenger comfort can be ambiguous to define, so we simply record the second to sixth derivatives of the position vector with respect to time, respectively termed acceleration, jerk, snap, crackle, and pop. In Table 1 we note the 99th percentile of each statistic given all data collected per path planning method. Generally speaking, lower numbers correspond to a smoother driving experience.

Table 1: We evaluate different path planning methods based on two CARLA environments: `Town01`, which each method was trained on; and `Town02`: a test environment.

| Town01 | Successes | Collision Impulse | Wrong lane | Off road | Accel | Jerk | Snap | Crackle | Pop |
|---|---|---|---|---|---|---|---|---|---|
| PID Controller | 0 / 10 | 8.92 | 18.6% | 12.1% | 0.153 | 0.925 | 9.19 | 85.8 | 785 |
| Cond. IL, PID Controller | 5 / 10 | 1.28 | 0.2% | 0.32% | 0.060 | 0.313 | 2.52 | 17.4 | 169 |
| Cond. IL, Learned Actions (Codevilla et al., 2018) | 7 / 10 | 0.96 | 9.8% | 1.64% | 0.203 | 0.674 | 5.52 | 46.9 | 438 |
| Model-Based RL (LaValle, 2006) | **10 / 10** | **0.00** | 9.3% | 0.82% | 0.062 | 0.353 | 2.69 | 26.1 | 261 |
| *Our method* | **10 / 10** | **0.00** | **0.0%** | **0.00%** | **0.054** | **0.256** | **1.50** | **13.8** | **136** |
| Town02 | Successes | Collision Impulse | Wrong lane | Off road | Accel | Jerk | Snap | Crackle | Pop |
| PID Controller | 2 / 10 | 12.5 | 5.0% | 4.99% | 0.204 | 1.040 | 6.77 | 59.1 | 611 |
| Cond. IL, PID Controller | 2 / 10 | 8.87 | 2.2% | 1.03% | 0.319 | 0.798 | 3.66 | 33.3 | 319 |
| Cond. IL, Learned Actions (Codevilla et al., 2018) | 3 / 10 | 1.23 | 21.6% | 3.06% | 0.368 | 1.234 | 8.13 | 91.1 | 845 |
| Model-Based RL (LaValle, 2006) | 7 / 10 | 2.56 | 12.0% | 3.53% | 0.134 | 0.967 | 6.06 | 63.1 | 575 |
| *Our method* | **8 / 10** | **0.41** | **0.4%** | **0.27%** | **0.054** | **0.613** | **2.64** | **21.4** | **289** |

The poor performance of the PID baseline indicates that the high-level waypoints do not communicate sufficient information about the correct driving direction. Imitation learning achieves better levels of comfort than MBRL, but exhibits substantially worse generalization from the training data, since it does not reason about the sequential structure in the task. Model-based RL succeeds on most of the trials in the training environment, but exhibits worse generalization. Notably, it also scores much worse than IL in terms of staying in the right lane and maintaining comfort, which is consistent with our hypothesis: it is able to achieve the desired goals, but does not capture the behaviors

Table 2: Incorporating a pothole cost enables our method to avoid potholes

| Approach | Successes | Pothole hits | Wrong lane | Off road |
|---|---|---|---|---|
| Our method without pothole cost, `Town01` | 9 / 10 | 177/230 | 0.06% | 0.00% |
| Our method with pothole cost, `Town01` | 9 / 10 | **10/230** | 1.53% | 0.06% |
| Our method without pothole cost, `Town02` | 8 / 10 | 82/154 | 1.03% | 0.30% |
| Our method with pothole cost, `Town02` | 7 / 10 | **35/154** | 1.53% | 0.11% |

in the data. Our method performs the best under all metrics, far exceeding the success and comfort metrics of imitation learning, and far exceeding the lane-obeyance and comfort metrics of MBRL.

## 4.1 Avoiding novel obstacles at test-time

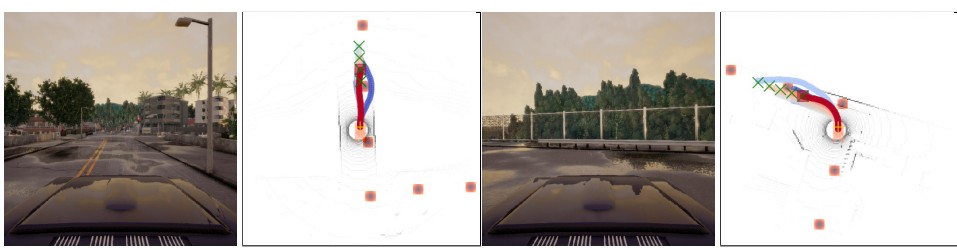

Figure 7: Test-time pothole planning. The preferred plans steer left around the simulated potholes.

To further illustrate the capability of our method to incorporate test-time costs, we designed a pothole collision experiment. We simulated 2m-wide potholes in the environment by randomly inserting them in the cost map offset from each waypoint, distributed $\mathcal{N}(\mu = [-15\text{m}, 2\text{m}], \Sigma = \text{diag}([1, 0.01]))$, (*i.e.* the mean is centered on the right side of the lane 15m before each waypoint). We ran our method that incorporates a test-time cost map of the simulated potholes, and compared to our method that did not incorporate the cost map (and thus had no incentive to avoid potholes). In addition to the other metrics, we recorded the number of collisions with potholes. In Table 2, we see that our method with cost incorporated achieved nearly perfect pothole avoidance, while still avoiding collisions with the environment. To do so, it drove closer to the centerline, and occasionally dipped into the opposite lane. Our model internalized obstacle avoidance by staying on the road, and demonstrated its flexibility to obstacles not observed during training. Fig. 7 shows an example of this behavior.

## 4.2 Robustness to poor-quality waypoints

As another test of our model's capability to stay in the distribution of demonstrated behavior, we designed a "decoy waypoints" experiment, in which *half* of the waypoints are highly perturbed versions of the other half, serving as distractions for our planner. The planner is tasked with planning to all of the waypoints under the Gaussian mixture likelihood. The perturbation distribution is $\mathcal{N}(0, \sigma = 8m)$: each waypoint is perturbed with a standard deviation of 8 meters. We observed the imitative model to be surprisingly robust to decoy waypoints. Examples of this robustness are shown in Fig. 8. One failure mode of this approach is when decoy waypoints lie on a valid off-route path at intersections, which temporarily confuses the planner about the best route. In Table 3, we report the success rate and the mean number of planning rounds for successful and failed episodes. These numbers indicate our method can execute dozens to hundreds of planning rounds without decoy waypoints derailing it.

We also designed an experiment to test our method under systemic bias in the route planner. Our method is provided waypoints on the wrong side of the road. We model this by increasing the goal likelihood observation noise $\epsilon$. After tuning the noise, we found our method to still be very effective at navigating, and report results in Table 3. This further illustrates our method's tendency to stay near the distribution of expert behavior, as our expert never drove on the wrong side of the road.

Table 3: Our method is able to ignore decoy waypoints in most planning rounds: it can execute dozens to hundreds of planning rounds without decoy waypoints derailing it. Our method is also robust to waypoints on the wrong side of the road.

| Approach | Successes | Avg. #plans until success | Avg. #plans until failure |
|---|---|---|---|
| Our method with $1/2$ waypoints noisy, Town01 | 4 / 10 | 157.6 | 37.9 |
| Our method with $1/2$ waypoints noisy, Town02 | 5 / 10 | 78.0 | 32.1 |
| Approach | Successes | Wrong lane | Off road |
| Our method with waypoints on wrong side, Town01 | 10 / 10 | 0.338% | 0.002% |
| Our method with waypoints on wrong side, Town02 | 7 / 10 | 3.159% | 0.044% |

## 5 Discussion

We proposed a method that combines elements of imitation learning and model-based reinforcement learning (MBRL). Our method first learns what preferred behavior is by fitting a probabilistic model to the distribution of expert demonstrations at training time, and then plans paths to achieve user-specified goals at test time while maintaining high probability under this distribution. We demonstrated several advantages and applications of our algorithm in autonomous driving scenarios. In the context of MBRL, our method mitigates the distributional drift issue by explicitly preferring plans that stay close to the expert demonstration data. This implicitly allows our method to enforce basic safety properties: in contrast to MBRL, which requires negative examples to understand the potential for adverse outcomes (e.g., crashes), our method automatically avoids such outcomes specifically because they do not occur (or rarely occur) in the training data. In the context of imitation learning, our method provides a flexible, safe way to generalize to new goals by planning, compared to prior work on black-box, model-free conditional imitation learning. Our algorithm produces an explicit plan within the distribution of preferred behavior accompanied with a score: the former offers interpretability, and the latter provides an estimate of the feasibility of the plan. We believe our method is broadly applicable in settings where expert demonstrations are available, flexibility to new situations is demanded, and safety is critical.

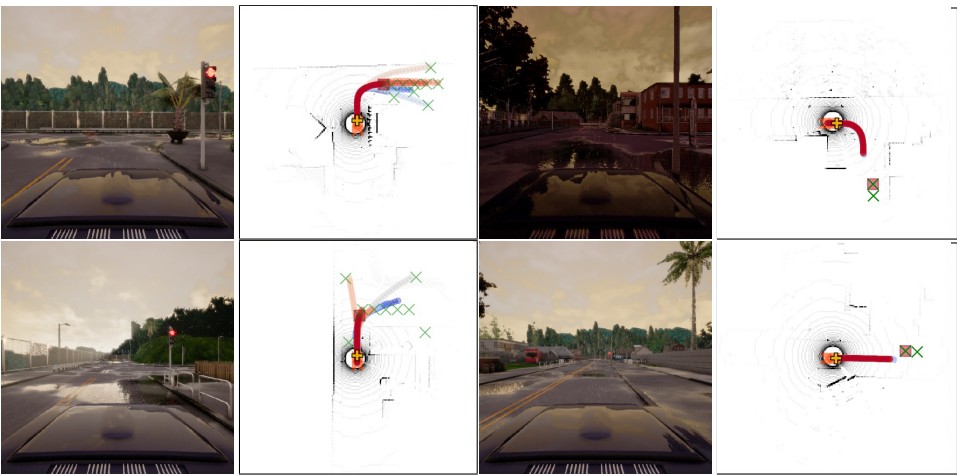

Figure 8: Tolerating bad waypoints. The planner prefers waypoints in the distribution of expert behavior: on the road at a reasonable distance. *Columns 1,2*: Planning with $1/2$ decoy waypoints. *Columns 3,4*: Planning with all waypoints on the wrong side of the road.

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
