# OpenReview forum: "Deep Imitative Models for Flexible Inference, Planning, and Control"
_ICLR.cc/2019/Conference_

### Official Review · AnonReviewer3 · 2018-11-02
**Good paper with detailed experiments, but the idea seems lacking novelty**

**Rating:** 6
**Confidence:** 1

**Review:**

Major Contribution:
The paper introduces a method that combines the advantage and of model-based RL and imitation learning and offset their weakness. The method proposes a probabilistic inference approach to analyze the action of the model.

Organization/Style:
The paper is well written, organized, and clear on most points.

Technical Accuracy:
I'm not an expert in RL. The method is obscure to me, but from my point of view, the experiments are done quite thoroughly and the results look good.

Presentation:
Good.

Adequacy of Citations:
The author should consider adding the related works include:
Bojarski, Mariusz, et al. "End to end learning for self-driving cars.": using CNNs to implement imitation learning for self-driving cars

Multimedia:
Videos are helpful to understand the method and are well composed.

---

> ### Author Response · Authors · 2018-11-14
> **relevant citation, contribution clarification**
>
> Thank you for your helpful feedback.
>
> Q1: "The author should consider adding the related works"
> A1: We have included your suggested reference.
>
> Q2: "Good paper with detailed experiments, but the idea seems lacking novelty"
> A2: We have since given more evidence of the novelty of our method with additional experiments in a revision. Our main contribution is a novel hybridization of model-based RL and Imitation Learning. This enables high performance in CARLA without any trial-and-error learning, and additionally enables flexibility to tasks not observed in the training data, such as avoiding potholes (Table 2, Figure 7) and robust navigation in the presence of noisy goals (Table 3, Figure 8).

---

### Official Review · AnonReviewer1 · 2018-11-03
**[Update] Elegant probabilistic formulation with limited experimental validation**

**Rating:** 6
**Confidence:** 3

**Review:**

# Summary

This submission proposes a method to combine the benefits of model-based RL and Imitation Learning (IL) for navigation tasks. The key idea is to i) learn a prior over trajectory distributions from a fixed dataset of demonstrations, and ii) use this learned dynamical model for path planning via probabilistic inference. Reaching target waypoints is done by maximizing the trajectory likelihood conditioned on the planning goal. The prior is learned using R2P2 on LIDAR features and past positions. Experiments using the CARLA driving simulator show that this method can outperform standard control, IL, and model-based RL baselines, while flexibly incorporating test-time goals and costs thanks to its probabilistic formulation.


# Strengths

The method is an elegant way to get the best of both worlds in RL and IL, leveraging the recent R2P2 work to estimate a powerful sequential model used for planning via probabilistic inference. The flexibility of the method in considering test-time cost maps and user-defined goals (e.g. to avoid potholes) is appealing, especially since it does not require on-policy data collection.

The proposed planning-as-inference method can in theory handle the multi-modality present in human demonstrations by using a probabilistic model of the observed behaviors as prior over undirected expert trajectories.

The approach seem to outperform both model-based and imitation learning baselines on a simplified version of the CARLA benchmark, including on interesting fine-grained metrics (e.g., comfort based).


# Weaknesses

The main weakness of this submission lies in its experimental evaluation, especially the absence of any dynamic objects in the tested environment ("static world CARLA", section 1). It is unclear how this approach would generalize beyond just staying on the road. How would it handle traffic lights, pedestrians, other drivers, weather variations, and more complex driving tasks than waypoint following by traversing mostly free space? How does the prior generalize to more complex behaviors (e.g, by using more contextual information \phi)? How robust is the method to noise in the demonstrations, i.e. non-expert or suboptimal behavior? It seems that estimating the generative prior on human behavior might suffer from the same issues as behavior cloning, e.g., the sample inefficiency due to the combinatorial explosion of causal factors explaining complex human behaviors. It might be in fact even harder to estimate that generative model than use a direct discriminative approach (e.g., a modular pipeline), at the cost of reduced flexibility at test time of course. The currently reported sample efficiency (7000 training samples) and near perfect success rate seem to suggest that this (non-standard) version of the CARLA benchmark is too simple (no weather variations, no dynamic obstacles). Comparison to the state of the art (beyond the baselines implemented here) on the original CARLA benchmark seems needed (especially in the "Nav. dynamic" task).

The method is only described very succinctly in section 2. I do not believe there are enough details (especially around the learning algorithm, hyper-parameters, and other important technical elements) for reproducibility at this stage. Section 2.1 is also quite dense for people not familiar with the R2P2 paper. As the main contribution of the paper is to leverage that model for planning and control, it would be great to maybe discuss a bit deeper. Finally, the input modalities are not clear, especially for the baselines: the proposed method is using LIDAR and localization whereas the IL baseline seems to use vision (while the others just use the trajectory). This makes the fairness of the comparison really unclear (LIDAR is a much stronger signal for just staying on the road).

Minor remarks:
- Why use a proportional controller as a baseline instead of the standard PID one?
- Section 2.3 seems like it's missing the extension of equation 2 to the multi-goal case?
- Typos in section 3 ("trail-and-error"), section 4 ("autonmous", "knowledge to")


# Recommendation

Although the theoretical benefits of the method are well-motivated and clear (off-policy learning, probabilistic model, flexibility at test time), the experimental evaluation (custom simple CARLA test, unclear comparison to baselines) and lack of details impeding reproducibility seems to suggest that this submission needs a bit more work. First, adding more details as suggested above and clarifying the experimental protocol seem like a must, but can be easily addressed by an update to the text. Second, it would be ideal to evaluate the approach on the standard CARLA benchmark in order to compare fairly to the prior art. This is much more involved.

I personally like the approach, so although I think it is marginally below the acceptance threshold in its current form, I reserve my judgement for the time being and look forward to the authors' reply.


# Update

The submission has been drastically rewritten (the diff is massive) and I think it is in much better shape, answering some of my concerns around reproducibility and generalization. Furthermore, it reinforces the strengths of the approach (esp. around its flexibility).

I am willing to recommend acceptance, but I have some further questions (hence I have only updated my score to a 6 for now). They are mostly related to the comparison with IL (important to validate the claim in the paper that the proposed approach is quantitatively better than both existing IL and RL methods). See discussion below for details.

---

> ### Author Response · Authors · 2018-11-14
> **generalization experiments, SOA comparison, method and baseline clarifications**
>
> Thank you for your helpful feedback.
>
> Q1: “Unclear how this approach would generalize beyond just staying on the road”
> We agree that there are more sophisticated settings that could be used to test different generalization aspects of our method. In theory, expert behaviors could be modelled in such settings by including the relevant information in the context, as noted by the reviewer. However, we designed our original experiments to reduce the number of uncontrolled variables, in order to clearly isolate the benefits of our approach. In order to test other generalization capabilities, we have since conducted additional experiments in several settings: obstacles in the road that were unseen in the demonstrations (i.e. simulated potholes), and noise in the waypoints provided to the controller, which could occur in a real-world setting due to noisy localization. In the pothole experiment, we found that our model was able to navigate around simulated potholes by including them in the cost map, and compared it to our model that was not provided with a cost map of the potholes. This navigation demands the model generalize its planning to situations not observed in the training data, specifically, when the car must partially enter the opposing lane in order to avoid the obstacle. In the noisy waypoint experiment, we tested two different types of noise: high bias, low variance noise, and low bias, high variance noise. In the high variance setting, “decoy” waypoints are added to the set of possible waypoints. The decoys are obtained by significantly perturbing the original waypoints with Gaussian noise, sigma=8 meters. Successful navigation in this setting required the model’s ability to score its plans by  likeliest under the estimated expert’s distribution of behavior. In the high bias setting, all waypoints were provided on the wrong side of the road, which is modelled with a small amount of observation noise. We found that these waypoints were still sufficient to communicate high-level navigation directions, and that the model usually produced plans on the correct side of the road (where all expert demonstrations occurred). Please see the updated results for quantitative (Tables 2, 3) and qualitative comparisons (Figures 7, 8).
>
> Q2: “Comparison to the state of the art (beyond the baselines implemented here) seems needed”
> A2: Our problem motivation is, instead, that of completely offline learning, but state-of-the-art CARLA results require trial-and-error based data collection online (Codevilla, et al. 2018). Additionally, navigation performance isn’t the sole goal of our method; we also show that our model has flexibility to different test-time queries that require behavior not seen in the training data. However, we have since implemented the “branched” architecture of Codevilla, et al. 2018, and trained it with the same inputs and data used to train our method.  We found this approach to slightly outperform the original IL baseline we included in our paper, but underperform the MBRL comparison and our proposed method. Please see the updated results for our quantitative comparison (Table 1).
>
> Q3: “The method is only described very succinctly in section 2”
> We have included many more details about the method and the implementation in our updated version. Please see Section 2, and Section 2.2 in particular.
>
> Q4: “the input modalities are not clear, especially for the baselines”
> The input modalities are identical for all methods: they all receive the same waypoints, and observe the same LIDAR and past trajectory. We clarified this in the updated paper.
>
> Q5: “Why use a proportional controller as a baseline instead of the standard PID one?
> We tested added I+D terms, replacing the P-controller with a PID controller, and found no significant change -- the PID controller fundamentally cannot handle faraway waypoints.
>
> Q6: “Section 2.3 seems like it's missing the extension of equation 2 to the multi-goal case?”
> We have generalized the mathematical explanation, from which all of our inference procedures can be derived. This includes the multigoal case, in Section 2.1 in the updated version. Additionally, we’ve included a qualitative demonstration of planning to sequential multi-goals (Figure 3).

---

> > ### Comment · AnonReviewer1 · 2018-11-27
> > **Improved text and experiments, still some questions related to IL comparison**
> >
> > Thanks for the reply and revision. The submission has been drastically rewritten (the diff is massive) and I think it is in much better shape, answering some of my concerns around reproducibility and generalization. Furthermore, it reinforces the strengths of the approach (esp. around its flexibility).
> >
> > I am willing to recommend acceptance, but I have some further questions (hence I have only updated my score to a 6 for now). They are mostly related to the comparison with IL (important to validate the claim in the paper that the proposed approach is quantitatively better than both existing IL and RL methods).
> >
> > 1) "state-of-the-art CARLA results require trial-and-error based data collection online (Codevilla, et al. 2018)"
> > What do you mean? CIL is a behavior cloning approach that is trained off-line? Are you talking about the data augmentation / noise injection?
> >
> > 2) Why are the results of IL much worse than the RL ones while in past publications it's the opposite (cf. CoRL'17 CARLA paper for instance)? This is what I meant by using the "standard CARLA benchmark" for the navigation part of the experiments: you would not just compare to baselines or reimplementations. If the results are counter-intuitive or in (apparent) opposition with previous peer-reviewed and published results, then the submission falls in the "extraordinary claims require extraordinary evidence" regime. And in this case, it is still not clear to me why your IL baselines are that low, and what makes your method not testable on the CARLA benchmark for the navigation tasks. This is not a huge deal breaker, because the proposed method has other clear advantages over IL and RL, but this still casts a shadow on the quantitative comparison with IL.
> >
> > 3) Why using LIDAR as input and not the images as most related works on CARLA? Your method seems to be directly applicable to image inputs (\phi contains a HxWxC tensor). I understand the main benefit of LIDAR is it provides a much stronger signal for collision avoidance (esp. for dynamic objects), but I would like to see how this approach works in the more common case of image inputs (and this is obviously linked to question 2 above).

---

> > > ### Author Response · Authors · 2018-11-27
> > > **IL comparisons**
> > >
> > > Thank you for your reply.
> > >
> > > Q1:
> > > Yes, we refer to the noise injection, used for better generalization and avoiding "unstable policies" (Codevilla, et al. 2018). Our reasoning is injected noise constitutes an intervention taken by the machine. Consequently, training could be more dangerous, especially at speed: humans experts are not used to driving with injective noise, and would possibly require special training prior to data collection. An additional requirement is vehicle modifications to actuate such signals. We appreciate it is ambiguous if "injected noise" constitutes "trial-and-error online" (or not), and clarify it constitutes ‘intervention’. In a related paper, (Liang, et al. 2018) solve the generalization issue by using trial-and-error DDPG after imitation learning. Our paper differs from both (Codevilla, et al. 2018) and (Liang, et al. 2018) in that we require no interventions of any sort (injected noise, nor explicit trial and error). Our method instead solves the imitation generalization issue a new way: by explicitly and probabilistically modelling multi-step expert trajectories.
> > >
> > > Q2:
> > > One source of difference is that we used Model-based RL, instead of Model-free RL (which is used in the CARLA CoRL 2017 paper). The Model-Based RL baseline is strong because of its access to the LIDAR map, which provides very useful obstacle cues. Therefore, we should not necessarily expect the same relative performances between IL and RL. Another source of difference is because we use LIDAR in all methods, and not vision (explained further in next question), a clear comparison with the previous vision-based IL benchmark was difficult. We judged a more meaningful comparison was for all baselines to use the exact same input as our method (LIDAR and previous vehicle locations). To do this, we reimplemented the IL baseline for LIDAR input. Given the significant difference between LIDAR and vision, we do not believe the IL baseline performance is suspiciously low. Our experiments fall under the "Navigation" CARLA benchmark, with the sole difference that our setting is always harder: goal locations are always placed further away (we selected goal destinations as the furthest distance from the starting position) for consistency difficulty across trials. Given the significant difference between LIDAR and vision, we will make more explicit in our paper's conclusions that our findings are specific to LIDAR-input cases only.
> > >
> > > Q3:
> > > Whilst vision would certainty be an interesting extension to this work, we used LIDAR for the following reasons. We used the R2P2 RNN method from (Rhinehart et al. 2018) for our imitative model q, which makes use of the LIDAR's overhead representation of the scene to build an overhead spatial cost grid of the scene, with the same 200x200 size as the lidar input (discussed briefly Section 2.2 of updated paper, which we will expand and clarify). This overhead representation helps to learn a spatial cost map, and is in the same 2D Euclidean space that our predictive model reasons about when predicts trajectories, making LIDAR a natural input for out method to use. To use vision as \phi, our method would need to transfer to an overhead representation, which is nontrivial. We agree vision would certainly be a worthwhile extension but is not the focus of this work. We will additionally clarify these reasons for why we use LIDAR in the paper.

---

### Official Review · AnonReviewer2 · 2018-11-05
**This paper combines Imitation Learning (IL) and Model Base Reinforcement Learning (RL) to come up with a novel algorithm that can take in user-defined targets while maintaining expert like behaviors.   This  promising approach that combines the benefits of IL and RL but with result performed only in simulation.**

**Rating:** 5
**Confidence:** 5

**Review:**

- Does the paper present substantively new ideas or explore an under explored or highly novel
question?

Yes, the paper combines two frameworks (Imitation Learning and Model Base
Reinforcement Learning) to incorporate target information while fitting to the expert distribution. Maybe, the idea is novel but experiments are only in simulation.

- Does the results substantively advance the state of the art?

 No, the compared methods are not state-of-the-art.

-  Will a substantial fraction of the ICLR attendees be interested in reading this paper?

 Yes a substantial fraction of ICLR attendees might be interested in reading the paper.

 - would I send this paper to one of my colleagues to read?

Yes.


- Quality:

The key point of this paper is that the proposed algorithm is novel and combines
the advantages of Imitation Learning and Model Base Reinforcement Learning. However, the
authors do not address the problem of IL when the stochasticity in the environment and/or model
results in trajectories outside of expert’s distribution. Additionally, all experiments are done in
simulation only and comparisons are made against components of the proposed algorithm instead
of the state-of-the-art.  This is definitely a limitation of the paper given recent works on imitation learning and model predictive control  as applied to real robotic systems in the task of agile off-road visual navigation.

In addition, the paper does not provide any detail on the training procedure (Network architecture, cost
function, etc), which makes results hard to reproduce. In addition, the experiments only compare
the proposed algorithm to its components, namely proportional controller, IL only controller and
Model Basel RL only controller.

- Clarity:

Easy to read. Thorough comparison with existing frameworks (Advantages compared to IL and model
based RL).

Originality:

– Novel algorithm presented with success in simulation.

---

> ### Author Response · Authors · 2018-11-14
> **generalization, new experiments, updated details for reproducibility**
>
> Thank you for your helpful feedback.
>
> Q1: “The authors do not address the problem of IL when the stochasticity in the environment and/or model results in trajectories outside of expert’s distribution.”
> A1: In our original submission, we evaluated our model’s ability to control the agent in a held out test scene (Town02). This demonstrated our model’s ability to generalize its behavior beyond the behaviors observed in the data. As further evidence of generalization, we performed additional experiments designed to force the model to produce trajectories outside of the distribution of observed trajectories. In one, we added simulated potholes to the scene, which we modelled with a cost map. This forced our planning to produce trajectories that avoid the potholes. We found that the model could still complete most of its episodes, while avoiding most potholes, despite the fact that the agent was forced into situations not seen in the training data. Please see the revised paper for these results.
>
> Q2: “the experiments only compare the proposed algorithm to its components, namely proportional controller, IL only controller and Model Basel RL only controller.”
> A2: We agree that relevant comparison is important. Our current IL comparison is not an ablation of our method, but rather a comparison to prior offline IL work. It most closely resembles the method of Codevilla, et al. "End-to-end driving via conditional imitation learning." ICRA, 2018. However, this prior method uses categorical command prediction, "turn left/turn right/go straight", for a learned lower-level controller, whereas our variant of this method regresses setpoints provided to a PID controller. We did not make the connection clear in the original paper, which we will fix.
> We also conducted additional experiments against the state-of-the-art with the “branched” network of Codevilla, et al. 2018, which we include in our revised comparison. We found this approach to slightly outperform the original IL baseline we included in our paper, but still underperform the MBRL method and our proposed method. Please see the updated paper for our quantitative comparison.
>
> Q3: “the paper does not provide any detail on the training procedure (Network architecture, cost function, etc), which makes results hard to reproduce”
> A3: In our updated version, we have simplified our explanation and expanded on additional details, including network architecture, cost function, etc. Please see Section 2.2 in the updated paper.

---

### Meta-Review · Area_Chair1 · 2018-12-12
**RL methods anno 2018**

**Confidence:** 4
**Recommendation:** Reject

**Metareview:**

This paper proposes to combine RL and imitation learning, and the proposed approach seems convincing.

As is typical in RL work, the evaluation of the method is not strong enough to convince the reviewers.  Increasing community criticism on RL methods not scaling must be taken seriously here, despite the authors' disagreement.